# The Second Human Pegivirus, a Non-Pathogenic RNA Virus with Low Prevalence and Minimal Genetic Diversity

**DOI:** 10.3390/v14091844

**Published:** 2022-08-23

**Authors:** Shuyi Chen, Haiying Wang, Emmanuel Enoch Dzakah, Farooq Rashid, Jufang Wang, Shixing Tang

**Affiliations:** 1Department of Epidemiology, School of Public Health, Southern Medical University, Guangzhou 510515, China; 2Department of Molecular Biology and Biotechnology, School of Biological Sciences, College of Agriculture and Natural Sciences, University of Cape Coast, Cape Coast CC 033, Ghana; 3School of Biology and Biological Engineering, South China University of Technology, Guangzhou 510006, China

**Keywords:** second human pegivirus, genome fidelity, error-prone RNA polymerase, selection pressure, attenuated viral vaccines

## Abstract

*The second human pegivirus* (HPgV-2) is a virus discovered in the plasma of a *hepatitis C virus* (HCV)-infected patient in 2015 belonging to the *pegiviruses* of the family *Flaviviridae*. HPgV-2 has been proved to be epidemiologically associated with and structurally similar to HCV but unrelated to HCV disease and non-pathogenic, but its natural history and tissue tropism remain unclear. HPgV-2 is a unique RNA virus sharing the features of HCV and the first human pegivirus (HPgV-1 or GBV-C). Moreover, distinct from most RNA viruses such as HCV, HPgV-1 and human immunodeficiency virus (HIV), HPgV-2 exhibits much lower genomic diversity, with a high global sequence identity ranging from 93.5 to 97.5% and significantly lower intra-host variation than HCV. The mechanisms underlying the conservation of the HPgV-2 genome are not clear but may include efficient innate immune responses, low immune selection pressure and, possibly, the unique features of the viral RNA-dependent RNA polymerase (RdRP). In this review, we summarize the prevalence, pathogenicity and genetic diversity of HPgV-2 and discuss the possible reasons for the uniformity of its genome sequence, which should elucidate the implications of RNA virus fidelity for attenuated viral vaccines.

## 1. Introduction

Only two human *pegiviruses*, the first human pegivirus (HPgV-1/GBV-C) and the second human pegivirus (HPgV-2), have been found to exist in the family *Flaviviridae*. Many studies on HPgV-1 have been conducted since its discovery in 1995 [1]. HPgV-1 is transmitted sexually, parenterally and by mother-to-child transmission, with high prevalence exceeding 40% in populations at high risk for exposure to bloodborne agents [2,3]. HPgV-1 commonly coinfects with hepatitis C virus (HCV) and human immunodeficiency virus (HIV), but is non-pathogenic. HPgV-1 is considered to be a “good virus” because it modulates infection and disease progression in HIV [3,4,5,6], and it improves the survival of Ebola patients when it coinfects with the Ebola virus [7].

HPgV-2 was firstly reported by Kapoor A. [8] and Berg M.G. [9] independently in 2015. Kapoor A. identified the virus in serum samples from 2 of 46 blood-transfusion recipients and 2 of 106 hemophilia patients, and they named the virus human hepegivirus 1 (HHpgV-1) due to its sequence similarity to both *hepaciviruses* and *pegiviruses* [8]. Berg M.G. identified the virus in an HCV-infected patient with multiple bloodborne exposures, who died from sepsis of unknown etiology, and named the virus HPgV-2 [9]. Successively, in 2016, Bonsall D. identified HPgV-2 from 2 of 150 HCV-infected patients and 1 of 195 hemophilia patients [10]. In 2017, Wang H. detected 9 HPgV-2-positive cases out of 70 HCV/HIV-1-coinfected patients who inject drugs (PWIDs) in Guangdong province of China and 43 positives out of 270 PWIDs in Sichuan province [11]. Compared to HPgV-1 and HCV, HPgV-2 is detected at a low frequency, and HPgV-2 seems to be a close relative of HCV. Almost all the *pegiviruses* are able to infect their hosts persistently without apparent pathogenicity. To date, only one *pegivirus*, an equine pegivirus (EPgV-TDAV) identified in 2013, has been shown to cause an acute hepatitis called Theiler’s disease in horses [12]. According to current studies, HPgV-2 has been able to establish a chronic infection in the presence of antibody responses for several years, but no evidence has indicated that it can cause disease in humans or that it is related to liver injury in the way HCV is [8,11,13,14]. The natural history, pathogenicity, tropism, relationship with HCV infection and potential impact on human health of HPgV-2 remain to be determined.

Although only ~36 full-length HPgV-2 isolates have been reported to date, it is established that the various HPgV-2 strains show high genome sequence identity, as the 11 HPgV-2 strains from the United States reported by Berg M.G show an overall sequence identity of 94~96% [9] while the 6 strains from China reported by Wang H. exhibited an identity of 93.6~97.8% [11]. Further studies have indicated that the HPgV-2 genome sequence seems to exhibit minimal diversity independent of geographical and temporal factors [15]. In 2018, four HPgV-2 strains from Iran were reported, two of whose genome sequences showed 99.6% identity to those of AK-790 strains from the United Kingdom, and the other two showed 100% identity to IDU31 and HCV121 strains from China [11,16,17]. Two Cameroonian HPgV-2 strain sequences showed 93–94% identity to other HPgV-2 strains [18]. Comparatively, HCV and HPgV-1 are highly diverse and grouped into seven major genotypes and numerous subtypes, showing distinct differences in geographic distribution [19,20,21,22].

Whether HPgV-2 possesses low sequence diversity and biological characteristics distinct from those of other RNA viruses is of great interest. This review summarizes the existing knowledge on the prevalence, pathogenicity and genetic diversity of HPgV-2 and discusses how the virus maintains such a high genomic fidelity, which should provide insight into the genetic diversity and evolution of RNA viruses, potentially revealing the implications of RNA virus fidelity for live attenuated viral vaccines.

## 2. Prevalence of HPgV-2

The distribution and prevalence of HPgV-2 infections worldwide remain to be determined. Generally, HPgV-2 seems to have a low global prevalence in the HCV-infected and hemophiliac populations, and it hardly infects healthy people. In the last decade, research from different countries indicated that HPgV-2 infections rarely happen in the healthy population and the HBV- and HIV-monoinfected populations (seroprevalence less than 1%), but are closely associated with HCV and HCV/HIV coinfection (Table 1). Specifically, the positivity rate for HPgV-2 RNA in HCV-infected patients was 0.29% (7/2440) in China and 1.14% (23/2007) in the United States, and positivity for HPgV-2 antibodies was slightly higher, at 1.23% (30/2440) in China and 2.44% (49/2007) in the United States [8,9,15,17]. HPgV-2 seems to have a higher prevalence in the HCV/HIV-1-coinfected population, with RNA and antibody positivity rates of 3.47% (7/202) and 8.91% (18/202), respectively, in China, and a 6.3% (5/79) HPgV-2 RNA positivity rate in Vietnam [14,17]. Surprisingly, none of the 30 HIV-1-infected and 36 HCV-/HIV-1-coinfected PWIDs in the United Kingdom were positive for HPgV-2 RNA [10] (Table 1). The discordant results warrant more studies in different countries to address the association between HPgV-2 and HCV/HIV-1 coinfection. Moreover, a recent report from Coller K.E. et al. [23] provided evidence of HPgV-2 infections in 11.2% (22/197) of past or current HCV-infected injection drug users (IDUs) and 1.9% (4/205) of HCV-negative IDUs, indicating that HPgV-2 could establish infections and replicate in the absence of HCV, while HCV infection increases the risk of HPgV-2 infection. Therefore, it is concluded that HPgV-2 may infect and replicate independent of HCV, but HCV coinfection may significantly enhance HPgV-2 infection and replication (11.2% vs. 1.9% in Coller’s study).

## 3. Pathogenesis of HPgV-2

The pathogenesis and the role of HPgV-2 in HCV infection are still unclear. Though it is confirmed that HPgV-2 is closely related to HCV infection, there is no evidence that HPgV-2 contributes to liver disease. In 2017, Kandathil A.J. et al. detected 17 (10.9%) HPgV-2-positive cases out of 156 members of a cohort of persons who injected drugs, by quantitative PCR, among whom the median liver stiffness did not differ between 10 HPgV-2-positive versus 104 HPgV-2-negative cases [13]. Consistently, HPgV-2 infection did not increase the levels of the liver-specific enzyme alanine aminotransferase (ALT) in the HCV-infected blood donors, and the ALT levels were not significantly different between the HPgV-2-RNA-positive and negative subjects [17]. The tropism of HPgV-2 is unclear, but it does not appear to be linked to hepatitis, even though the virus was identified in the liver of one person by Kandathil A.J., which might have been due to contaminating blood in the liver [13,15,24]. *Pegiviruses* are known to infect lymphocytes. HPgV-1 is a lymphotropic virus causing persistent infection in both T and B lymphocytes and has proved to be closely associated with the overall risk of lymphoma [25,26,27]. Accordingly, the tropism of HPgV-2 also appears to be for lymphocytes. Only one detected case of HPgV-2 in an individual liver and HPgV-2 from patient plasma was unable to replicate in cultured hepatocyte cell lines [13,15]. Moreover, a recent study found that the virus’ RNA and non-structural antigen were detected in lymphocytes, while the positive- and negative-strand HPgV-2 RNA were detected in PBMCs, especially in B cells [28]. Therefore, HPgV-2 does not contribute to HCV induced liver injury. Moreover, someone wonders whether HPgV-2 has a beneficial effect for HIV disease progress, similar to HPgV-1. However, the relationship between HPgV-2 and HIV infection is still unclear and no evidence has shown that HPgV-2 infection has a positive impact on HIV disease.

HPgV-2 is able to establish a chronic infection with an antibody response. A longitudinal analysis of HPgV-2 was performed by Wang H. with 8 HPgV-2-infected patients in a period spanning 68 to 1843 days; most of the patients remained positive for anti-HPgV-2 antibodies but negative for HPgV-2 RNA during the follow-up period. Specifically, the patient HCV121 remained both anti-HPgV-2-antibody- and RNA-positive for about 2 years [17]. In another study, the median duration of HPgV-2 viremia was suggested to be at least 4538 days [13]. Furthermore, it has been shown that approximately 30% of HPgV-2 infections are persistent and the other 70% of infected subjects clear viremia during disease. Nevertheless, how HPgV-2 maintains a persistent infection and whether HPgV-2 stimulates adaptive and innate immune responses remains unclear. HPgV-2 infection might not stimulate robust adaptive and strong innate immune responses [29]. For the patient HCV121, after treatment with direct acting antivirals (DAAs), HCV RNA was eliminated, but the HPgV-2 RNA remained present, which was considered to represent HPgV-2 monoinfection. Under HPgV-2 monoinfection, proinflammatory cytokines were significantly downregulated, while cytokines regulating the innate immune response remained unchanged.

Further investigation is needed to reveal the mechanism of HPgV-2 pathogenesis and its relationship with HCV infection or HCV/HIV coinfection. Development of HPgV-2 cell culture systems and small animal infection model would be powerful tools and ideal research directions of HPgV-2.

## 4. HPgV-2 Genome Organization

HPgV-2 has a positive-sense, single-stranded RNA genome of approximately 9800 nucleotides with a single open reading frame (ORF) flanked by the 5′ and 3′ untranslated regions (UTRs). The ORF is translated into a multifunctional polyprotein of approximately 3000 amino acids containing the nucleocapsid protein (S), structural envelope glycoproteins (E1 and E2), unknown function protein X and non-structural proteins (NS2, NS3, NS4A/B and NS5A/B) (Figure 1). Sequence divergence between HPgV-2 and other *pegiviruses* is apparent; they show only 25.9–32.0% overall amino acid identity, and HPgV-2 shows 25.9% identity to HPgV-1 and 25.6% identity to HCV genotype 1 [8,9]. Phylogenetic analysis indicated that HPgV-2 strains from China, the United States and the United Kingdom clustered together to form a separate branch closely related to pegiviruses from bats and rodents [8,9,11,13,17].

Most *pegiviruses* contain 5′ UTRs with similar domain organization forming highly conserved internal ribosome entry site (IRES) motifs, named pegi-like IRESs, including motifs in domains IIIa/IIIb, at the apex of domain IVa, and in domain Vb. The HPgV-2 5′-UTR sequence is non-homologous to most of the other *pegiviruses*’ 5′ UTRs and contains a sequence highly conserved in the *hepacivirus* genus, TACAGCCTGATAGGGT, at position 274, which lies within domain IIIe of the *hepacivirus* IRES [8]. The structural alignment of the 5′ UTRs of HPgV-2 and *hepacivirus* revealed that HPgV-2 contains a typical type IV IRES but not a pegi-like IRES (Figure 1). The E2 protein of HPgV-2 contains 11 potential N-linked glycosylation sites and 1 potential O-linked site, which is much more than those of other pegiviruses but resembles those of HCV. Accordingly, in most HPgV-2 viremic persons (92.86%), like for HCV, both RNA and anti-E2 antibody can be detected, which is rare in HPgV-1 (5.88%) [21,24].

## 5. Minimum Genetic Diversity of HPgV-2

### 5.1. Conservation of HPgV-2 Genome

Evidence suggests that, unlike the genomes of other RNA viruses, the HPgV-2 genome exhibits high conservation, and all the HPgV-2 isolates group into one genotype. As shown in Figure 2 below, 54 HPgV-2 isolates’ genome sequences from the National Center for Biotechnology Information (NCBI) were used for the sequence diversity analysis of HPgV-2; of these isolates, 37, 5, 5, 4 and 2 were from the United States, Vietnam, China, Iran and the United Kingdom, respectively.

The comparison of the whole-genome sequences of the HPgV-2 isolates from different countries of origin with the consensus sequence revealed that the Vietnamese, UK, Iranian, Chinese and US isolates showed nucleotide sequence identities of 93.32–97.34% (Figure 2A). Additionally, 54 reported HPgV-1 sequences and 50 HCV sequences were downloaded from NCBI, and the nucleotide sequence identities were calculated. The 54 HPgV-1 isolates showed 82.10–85.12% sequence identity (Figure 2A), while the HCV isolates showed 69.11–84.67% identity, both of which showed far more diversity than HPgV-2. HPgV-2 exhibited high nucleotide (NT) and amino acid (AA) identities, with medians of 93.7% and 94.6%, respectively. Pairwise comparisons of the HPgV-2 strains showed high identities, with a median NT identity of 94.82% and median AA identity of 95.18%, while HPgV-1 had a much lower NT identity of 87.92% and a higher AA identity of 97.01% (Figure 2B). The HPgV-2 nucleotide sequence exhibits far less diversity than that of HCV and HPgV-1. Furthermore, the mutation of HPgV-2 amino acids was proved to be confined to a limited number of residues with 73% (2231/3057) of the AAs unchanged in the 34 strains analyzed [15]. Maximum likelihood phylogenetic analysis of HPgV-2, HPgV-1 and HCV was performed. HPgV-2 isolates from different countries clustered together to form a separate branch and fell into one group, while HPgV-1 and HCV segregated into several genotypes (Figure 3).

### 5.2. Stability of HPgV-2 Genome

The HPgV-2 genome sequence is also highly stable compared to the genome sequences of other RNA viruses, appearing to have fewer nucleotide changes over time. The sequences of the HPgV-2 strains from patient HCV121, at two time points about 2 years apart, showed 97% identity at the nucleotide level and 99% at the amino acid level [17]. Two HPgV-2 strains, ABT0035P. US and ABT0041P. US, were obtained from the same patient with a time interval of 7 weeks. This patient was also infected with relatively similar titers of HIV, HCV and HPgV-1; the numbers of nucleotide differences for HIV, HCV, HPgV-1 and HPgV-2 in the 7-week time frame were compared. Four methods were used to calculate the percent differences of the virus nucleotide sequences. Regardless of the method used, the percentage difference for HPgV-2 was lower than 0.1%, implying that the HPgV-2 sequence identity between two time points, 7 weeks apart, was consistent at >99.9%. Comparatively, the sequence identities of HIV, HCV and HPgV-1 across the 7 weeks were 98.75%, 99.72% and 99.25% [15]. Consequently, the sequence diversity over time for HIV, HCV and HPgV-1 was 10–25 times greater than that for HPgV-2. Taken together, these data suggest that HPgV-2 can persist chronically for months to years without any appreciable changes in sequence.

### 5.3. HPgV-2 Shows Less Intra-Host Diversity than HCV

The high mutation rates triggered by the error-prone viral RNA polymerase result in the formation of a large collection of closely related viral genomes around a master sequence, defined as viral quasispecies. Similar quasispecies exist for HPgV-2 as for HCV, but the intra-host genetic diversity for the HPgV-2 genome is much lower than that for the HCV genome. The intra-host single nucleotide variation (iSNV) analysis indicated that the iSNVs/kb for HPgV-2 ranged from 1.9 to −24.9, while those for HCV ranged from 30.77 to 69.23. Furthermore, the genome diversity rates were < 0.01 × 10^−2^ to 0.56 × 10^−2^ substitutions/site/year for HPgV-2 and 0.71 × 10^−2^ to 12.03 × 10^−2^ substitutions/site/year for HCV. We also found that 74% (568/768) of the iSNVs occurred at the third codon position in HPgV-2 ORFs, resulting in nucleotide transition and synonymous substitution. The distributions of iSNVs at the first and third codon positions of HPgV-2 were 17.7% and 74.0%, respectively, compared with the 39.0% and 48.8% for HCV. Furthermore, 80.5% of the HPgV-2 iSNVs cause nucleotide transitions rather than transversions, which finally leads to a much smaller ratio of nonsynonymous over synonymous substitutions (N/S) for HPgV-2 than HCV (0.32 vs. 1.41) [29].

Summarily, as stated above, evidence has preliminarily indicated the unique characteristics of the novel HPgV-2 with a highly conserved and stable genome sequence, which maintains a high identity rate independent of the virus’ geographical origin and time and shows low genome diversity. However, only ~36 HPgV-2 isolates with full-length sequence have been reported to date. More HPgV-2 strains need to be discovered for further exploration of HPgV-2’s sequence diversity and pathogenesis and to reveal its relationship with HCV infection.

## 6. Possible Explanation for HPgV-2’s Genome Fidelity

In cellular organisms, cellular DNA replication is mediated by a network of proteins, including DNA polymerase and accessory proteins. They ensure that the genome is reproduced faithfully, detect and repair erroneous and damaged bases, remove UV-induced pyrimidine dimers and rejoin double-stranded breaks; this is referred to as replication fidelity. Due to this excellent proofreading and repair system, cellular DNA replication exhibits relatively high fidelity, with an estimated error rate of 10^−9^ to 10^−11^ per nucleotide site per round of replication [30,31,32]. Comparatively, positive-stranded RNA viruses with genomes <20 knt do not have such proofreading and error repair systems [33], resulting in a relatively high estimated error rate of 10^−4^ to 10^−6^ per nucleotide site per round of replication [34,35]. The replication fidelity for the RNA genome is primarily determined by the virus-encoded RNA-dependent RNA polymerase (RdRP), a polymerase with low fidelity, which introduces mutations into the viral population during each cycle of RNA replication, continuously generating new viral variants defined as quasispecies [36]. Moreover, viruses have to overcome natural selection pressure, such as innate immune defenses, adaptive immune responses and certain antiviral drugs, to establish robust infections. The ability of viruses to escape from host immune responses and virulence correlates with their evolutionary capacity, defined as the mutation-selection equilibrium. Beneficial mutations increase to fixation under positive selection, while deleterious mutants are removed by negative/purifying selection. To some extent, selection shapes viral diversity. Viral diversity is largely a consequence of the synergy of mutations and selection.

Several studies have elucidated the uniformity of the HPgV-2 genome, but how it maintains such a high sequence fidelity remains unclear, and only one reported study [15] has investigated the mechanism underlying this fidelity. In Forberg’s study, the ability of HPgV-2 to persist for years without appreciable changes in sequence was attributed to the fitness of the virus, the rarity of infections and the high rate of clearance. They provided three possible explanations. Based on the analysis of the synonymous (S) and non-synonymous (N) mutations, which showed that the N mutations occurred less frequently than S mutations for HPgV-2, with a rather low median dN/dS of 0.132, they suggested that new mutations were potentially purged by purifying selection. Therefore, the first explanation is that HPgV-2 may be an ancient, genotypically stable virus having little room for improvement, with efficient purifying selection, and secondly, it may be well-adapted to its host and able to avoid immune responses. Consistent with the two explanations above, Liang et al. reported that the ratio of N/S for HPgV-2 was much smaller than that for HCV (0.32 vs. 1.41), suggesting efficient purifying selection for HPgV-2. Moreover, the expression levels of proinflammatory cytokines were compared between HPgV-2/HCV-coinfected patients with elimination of HCV and healthy blood donors; HPgV-2 infection alone did not appear to stimulate robust adaptive and strong innate immune responses [29]. Finally, given the reliance of HPgV-2 infection on HCV, HPgV-2 would certainly benefit from the coinfection. With the reliance on HCV, HPgV-2 may be under minimal selective pressure due to HCV reducing T-cell function and lessening T cells’ control over HPgV-2.

Besides viral fitness and selection, another possible mechanism is related to viral RdRP, which is critical for virus sequence diversity. An increase or decrease in the fidelity of RdRP results in changes in the replicate mutation rate, genetic diversity and, probably, virus fitness. The high-fidelity mutant of poliovirus (PV) G64S, with a single site 3D^pol^ Gly64 mutated to serine, shows an approximately three~five-fold decrease in the basal mutation rate compared to the wild-type virus and viral attenuation in vivo [37,38]. For another example, the high-fidelity influenza A virus (IAV) variant PB1-V43I, with an increased-fidelity RdRP, replicates to titers comparable to those for its wild-type counterpart in vitro or in mouse lungs, but shows reduced genetic diversity early post-infection and attenuated viral lethality (10-fold lower), with decreased viral neurovirulence [39]. Most of the reported high-fidelity mutants have shown fidelity increased by up to four- to five-fold compared to the wild-type counterpart and are often attenuated in vivo [37,40,41,42,43,44,45,46,47,48,49,50,51]. The higher-fidelity RdRP results in lower genetic diversity and virus attenuation [35,52,53,54]. However, there is no information about the biochemical or fidelity characteristics of HPgV-2 RdRP. The effect of HPgV-2’s RdRP on its genome diversity requires further evaluation, which might possibly require an available in vitro polymerase activity assay with RNA template for HPgV-2 RdRP.

## 7. Inspiration from HPgV-2

RNA viruses can attenuate themselves spontaneously by limiting genome diversity. Viruses with restricted diversity may be unable to overcome the selection pressure for failure in accumulation of mutations because of their low mutation frequencies, causing decreases in fitness and in vivo attenuation, which represents potential for developing attenuated live virus vaccines. The central point is how to achieve rational regulation of virus fidelity, and HPgV-2 will be an excellent study model. However, there is still a long way to go to achieve this goal because so little is known accurately about the HPgV-2. The high fidelity of HPgV-2 genome points to the future directions of the virus’ tropism, replication mechanism and RdRP fidelity, which might rely on the establishment of an in vitro and in vivo RNA polymerase assay, HPgv-2 cell culture system and small animal infection model. Furthermore, as aforementioned, viral attenuation and inefficient adaptation are the possible consequences of decrease in diversity, but it the association of non-pathogenesis and low prevalence of HPgV-2, with its minimal genetic diversity, remains to be elucidated. Revealing the underlying mechanism by which HPgV-2 maintains genome fidelity would not only expand our knowledge on RNA virus diversity and evolution but also establish a strong foundation for regulating virus fidelity to form stable attenuated viruses, providing new directions for antiviral strategies.

## Figures and Tables

**Figure 1 viruses-14-01844-f001:**
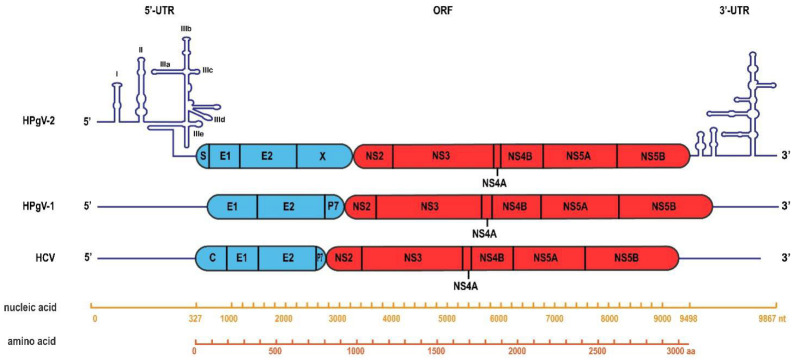
Genome organization of HPgV-2. The genome consists of the 5′ UTR, 3′ UTR and ORF region, encoding four structural proteins and six non-structural proteins [8,15].

**Figure 2 viruses-14-01844-f002:**
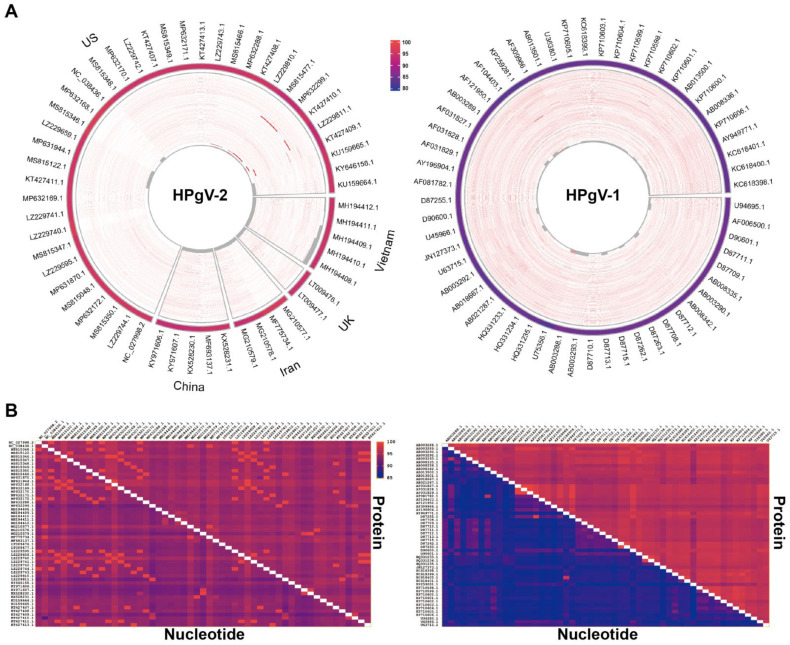
Nucleotide sequence diversity and pairwise alignment of HPgV-1 and HPgV-2 isolates with different countries of origin. (**A**) Whole-genome sequences of HPgV-2 and HPgV-2 from Vietnam, United Kingdom, Chinese and United States isolates were aligned with MAFFT multiple alignment and visualized with circles. The consensus sequence was generated with a 70% threshold. Mismatches are highlighted in red, and gaps are indicated in gray. The overall identities of each strain are indicated on a scale from 80% (blue) to 100% (red) in the peripheral ring. (**B**) Pairwise alignment of the nucleotide sequences and protein sequences of HPgV-1 (right panel) and HPgV-2 (left panel) strains. The heat map scale is from 85% (blue) to 100% (red).

**Figure 3 viruses-14-01844-f003:**
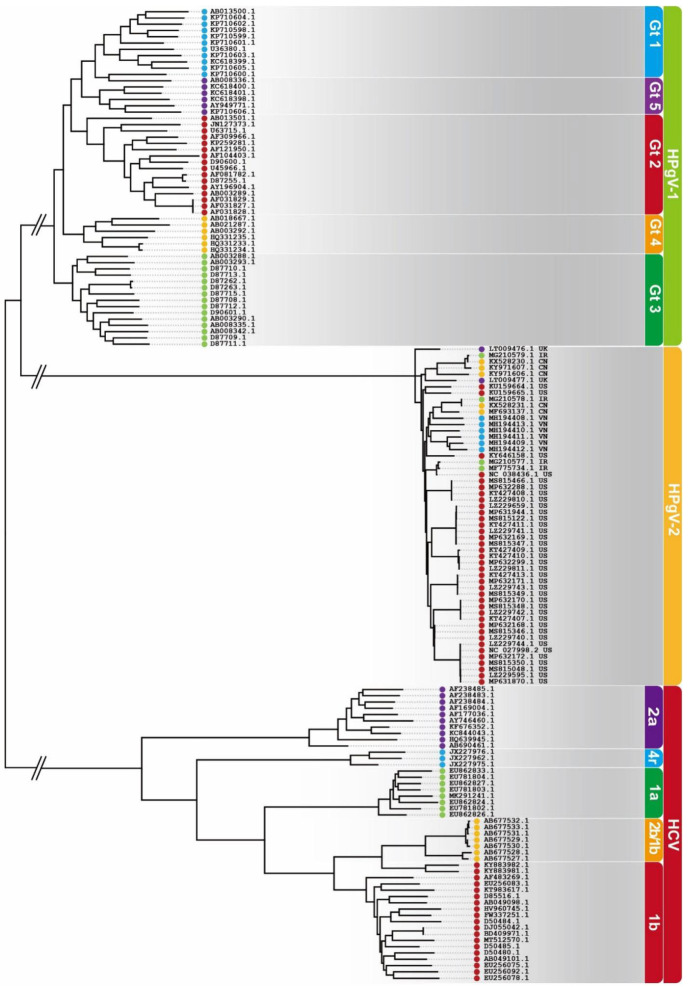
Phylogenetic trees of HPgV-1, HPgV-2 and HCV nucleotide sequences generated with the maximum likelihood method. The origins of HPgV-2 isolates are indicated as UK (●) for United Kingdom, IR (

) for Iran, CN (

) for China, US (

) for the United States and VN (

) for Vietnam.

**Table 1 viruses-14-01844-t001:** Serologic and molecular detection of HPgV-2 in different populations.

	Groups		No. Tested	Anti-HPgV-2 (+)	HPgV-2 RNA (+)
China	Healthy Blood Donors		4017	6 (0.15%)	0
	HBV (+)		1000	2 (0.20%)	0
	HIV-1 (+)		539	0	0
	HCV (+)		2440	30 (1.23%)	7 (0.29%)
	HCV (+)/HIV-1 (+)		202	18 (8.91%)	7 (3.47%)
	Total		8198	56 (0.68%)	14 (0.17%)
United States	Healthy Blood Donors		892	5 (0.56%)	0
	HBV (+)		944	8 (0.85%)	0
	HIV (+)		928	5 (0.54%)	0
	HCV (Ab+/NAT+)		1708	46 (2.69%)	22 (1.29%)
	HCV (Ab+/NAT-)		299	3 (1.00%)	1 (0.33%)
	Total		4771	67 (1.40%)	23 (0.48%)
Vietnam	Healthy Blood Donors		80	-	0
	HBV (+)		103	-	0
	HIV (+)		78	-	0
	HCV (+)		394	-	0
	HCV (+)/HIV-1 (+)		79	-	5 (6.3%)
	Total		734	-	5 (0.68%)
United Kingdom	Health Blood Donors		50	-	0
	Hemophilia		195	-	1 (0.5%)
	HIV (+)		36	-	0
	HIV (+)/HCV (+)/PWID		30	-	0
	HCV (+)/PWID		120	-	2 (1.7%)
	Total		431	-	3 (0.7%)
Iran	Hemophilia patients (*n* = 436)				
	HCV RNA (+)	HPgV-1 RNA (+)	TTV RNA (+)	HPgV-2 RNA (+)	HPgV-2 RNA (+)/HCV RNA (+)/HPgV-1 RNA (+)/TTV RNA (+)
	163 (37.4%)	19 (4.4%)	76 (17.6%)	4 (0.9%)	4 (0.9%)

## Data Availability

Data sharing is not applicable to this article, as no new data were created or analyzed in this study.

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
