# Peer review of "The Second Human Pegivirus, a Non-Pathogenic RNA Virus with Low Prevalence and Minimal Genetic Diversity"

_viruses, 2022, doi:10.3390/v14091844_

Round 1
Reviewer 1 Report
This is a review of HPgV-2, a human pegivirus discovered in 2015. Very little is known about this virus including its prevalence in various at-risk populations, global prevalence, pathogenic potential, and interactions with other viruses.
This review is poorly written and simply a listing of previously published facts. The writing is challenging to read and has multiple grammatical and typographical errors. It should be edited extensively by a native English speaker and/or a professional editing service.
There is no synthesis of the data. The authors do not explore “next steps” or suggest a comprehensive research agenda.
Line 57: how many full-length HPgV-2 isolates have been sequenced? How many other HPgV-2 isolates have been sequenced but not the complete genome? From what countries are full-length HPgV-2 genomes available?
Line 233: “ . . . only <100 HPgV-2 strains . . . “ is ambiguous as well.
Line 69: “We are of great interest . . .” is not correct English.
Are the previous reports only cross-sectional in nature or have longitudinal analysis of HPgV-2 RNA been conducted?
Detection of HPgV-2 should be described in more detail, preferably before sections on prevalence and pathogenesis. How is HPgV-2 detected? Only by detection of RNA? Targeting what genomic region? Or is there an antibody test too? Targeting what genomic region?
Line 129: “ . . . are still fuzzy.” is appropriate scientific language.
The order of Figures 2 and 3 should be reversed.
Figure 2A is difficult to understand. A more stand phylogram would be more informative in visualizing how distinct HPgV-2 sequences are from one another. The authors can then color the isolates by country of origin as in Figure 3.
For Figure 3, what do the colors for the HPgV-2 isolates correspond too? If country of origin, this is confusing since the country of origin is not given for the HPgV-1 or HCV isolates.
Section 7 is the most interesting part of this entire review but very short and largely uninformative.
Reviewer 2 Report
Chen, et al. reviewed the prevalence, pathogenicity and genetic diversity of HPgV-2, discuss the possible causes of genome sequence consistency. It will provide support for the research and development of HPgV-2 attenuated virus vaccine. The manuscripts is well prepared and the data presentation is evident to support the conclusion. The manuscript fits the scope of Virus.
Minor:Figure 2 resolution should be improved.
Reviewer 3 Report
The manuscript is well organized and comprehensively described. The author gives us a relatively complete introduction about the second human pegivirus (HPgV-2). In particular, the prevalence, pathogenesis, genome organization and minimal genetic diversity of hpgv-2 were introduced. These introductions can provide effective information for us to better understand the basic situation of the virus,And lay a good foundation for people to understand the pathogenic mechanism of the virus,It also provides relevant information for the development of effective vaccines. There is still a deficiency in the manuscript that needs to be improved,The latest research progress about relationship and pathogenicity of this virus with HIV and HCV infection,I think this is of great interest to many scientists, and it is more useful than summarizing a lot of literature.
Round 2
Reviewer 1 Report
This revised manuscript is improved; however, awkward phrases persist through the manuscript. Additionally, the comments below were not addressed adequately in the revision.
1) Original comment: There is no synthesis of the data. The authors do not explore “next steps” or suggest a comprehensive research agenda.
Response: Thanks for the comment. Revision has been made to address this point (line 145-148, line 314-315 in the revised manuscript).
New comment: Section 7 should be expanded significantly. What information about HPgV-2 is lacking? What studies should be done now? the authors must create a research agenda or next steps that shows a robust understanding of future studies / directions.
2) Original comment: For Figure 3, what do the colors for the HPgV-2 isolates correspond too? If country of origin, this is confusing since the country of origin is not given for the HPgV-1 or HCV isolates.
Response: Thanks to your comment. In Figure 3, the colors for the HPgV-2 isolates are corresponding to their origin, briefly, UK is indicated as purple, Iran is green, China is yellow, US is red and Vietnam is blue. The origin of the HPgV-2 isolates is also indicated behind the accession numbers as UK, IR, CN, US and VN.
New comment: The information in the response should be added to the legend for Figure 3.
3) Original comment: Section 7 is the most interesting part of this entire review but very short and largely uninformative.
Response: Thanks for your comment. Currently studies have shown the unique feature of HPgV-2 diversity which might open our eyes on the RNA virus evolution and vaccine development. But the mechanism of HPgV-2 genome fidelity modulation is almost totally unknown and studies on it are hardly reported. As we mentioned in the manuscript, it would be possibly related with special mechanism of immune escape or of RdRP fidelity, which are both needed to be confirmed by further studies. Our group has been trying to figure out how HPgV-2 maintain the genome fidelity, hoping that we can provide more information in the future.
New comment: This section must be expanded significantly.
Round 3
Reviewer 1 Report
The synthesis section is quite short. The authors could have expanded this in a more thoughtful and rigorous manner.